# Serine Racemase Deletion Affects the Excitatory/Inhibitory Balance of the Hippocampal CA1 Network

**DOI:** 10.3390/ijms21249447

**Published:** 2020-12-11

**Authors:** Eva Ploux, Valentine Bouet, Inna Radzishevsky, Herman Wolosker, Thomas Freret, Jean-Marie Billard

**Affiliations:** 1UNICAEN, INSERM, COMETE, Cyceron, CHU Caen, Normandie University, 14000 Caen, France; eva.ploux@unicaen.fr (E.P.); valentine.bouet@unicaen.fr (V.B.); thomas.freret@unicaen.fr (T.F.); 2Department of Biochemistry, Technion-Israel Institute of Technology, Rappaport Faculty of Medicine, Haifa 31096, Israel; vinna@technion.ac.il (I.R.); hwoloske@technion.ac.il (H.W.)

**Keywords:** d-serine, hippocampus, NMDA receptors, excitatory/inhibitory balance, synaptic plasticity, memory

## Abstract

d-serine is the major co-agonist of N-methyl-D-aspartate receptors (NMDAR) at CA3/CA1 hippocampal synapses, the activation of which drives long-term potentiation (LTP). The use of mice with targeted deletion of the serine racemase (SR) enzyme has been an important tool to uncover the physiological and pathological roles of D-serine. To date, some uncertainties remain regarding the direction of LTP changes in SR-knockout (SR-KO) mice, possibly reflecting differences in inhibitory GABAergic tone in the experimental paradigms used in the different studies. On the one hand, our extracellular recordings in hippocampal slices show that neither isolated NMDAR synaptic potentials nor LTP were altered in SR-KO mice. This was associated with a compensatory increase in hippocampal levels of glycine, another physiologic NMDAR co-agonist. SR-KO mice displayed no deficits in spatial learning, reference memory and cognitive flexibility. On the other hand, SR-KO mice showed a weaker LTP and a lower increase in NMDAR potentials compared to controls when GABA_A_ receptors were pharmacologically blocked. Our results indicate that depletion of endogenous D-serine caused a reduced inhibitory activity in CA1 hippocampal networks, altering the excitatory/inhibitory balance, which contributes to preserve functional plasticity at synapses and to maintain related cognitive abilities.

## 1. Introduction

d-serine is present at substantial levels in several brain areas, including the hippocampus [1,2,3,4,5], where it plays a major role in brain synaptic plasticity mechanisms involved in cognitive processes (reviewed in [6]). Experiments with brain slices perfused with the degrading enzymes d-amino acid oxidase or d-serine deaminase first uncovered the role of d-serine as the main co-agonist of the N-methyl-D-aspartate (NMDAR) subtype of glutamate receptors at the CA3/CA1 hippocampal synapses [7,8,9,10,11]. The NMDAR activation is the necessary step for the expression of functional plasticity in neuronal networks such as long-term potentiation (LTP) of synaptic transmission, a process involved in learning and memory [12,13,14,15,16,17]. In parallel, the role of d-serine was also investigated in brain slices from knockout mice with genetic deletion of the synthesizing enzyme serine racemase (SR) that converts L- into d-serine [18,19,20,21]. Genetic manipulations consisted of either the mutation of the SR gene in exon 9 (Srr^Y269^* mice, [22]) or the total deletion of exon 1 (SR-KO mice, [23,24]); both strategies reducing brain d-serine levels by at least 90%. Although several behavioral and electrophysiological changes have been reported in these transgenic mice (see [25]), a definitive picture is yet to emerge, presumably due to different experimental conditions employed. For example, there is an apparent discrepancy regarding changes in synaptic plasticity and NMDAR activation at CA3/CA1 hippocampal synapses. On one hand, studies of isolated excitatory glutamate transmission on CA1 pyramidal cells reported LTP deficits and altered NMDAR activation in slices from SR-KO mice [23,26,27]. On the other hand, extracellular recordings that monitor the overall network activity did not show differences in LTP [10,28]. We wondered if differences in the excitatory/inhibition (E/I) balance may underlie these apparent discrepancies. In order to investigate this possibility, we reinvestigated LTP expression and NMDAR synaptic activity with extracellular recordings in CA1 hippocampal networks of SR-KO mice before and after the alteration of the GABAergic inhibitory tone by the GABAR antagonist bicuculline. We found that SR deletion attenuates the inhibitory tone, which helps to preserve overall network activity and behavior.

## 2. Results

### 2.1. Cognitive Abilities Are Preserved in SR-KO Mice

Changes in locomotor activity (especially when considering transgenic animals) can deeply influence behavioral performances and generate bias of analyses and as such have to be controlled. Fortunately, the actimetry test did not reveal any difference in horizontal and vertical activities (traveled length and number of rearing) between the two genotype of mice groups (Figure 1A,B).

Besides, exploratory behavior appeared also unaltered in SR-KO mice. Indeed, we did not observe any between genotype differences, either for the number of transitions in the white and black box (Figure 1D) or the total traveled distance in the open field (Figure 1F).

Anxiety-related behavior was also assessed in the two genotype lines of mice. Thus, we did not show any group difference, as testified either by a similar latency to first entry into the black box (Figure 1C), the percentage of time spent in the white compartment (Figure 1E), or the percentage of time spent in the central zone of the open-field (Figure 1H). Thus, these results suggest similar anxiety-related levels between WT and SR-KO mice.

Unexpectedly, the percentage of time spent in the center during the first one minute was significantly higher in SR-KO mice (Figure 1I).

Learning and memory capacities were finally investigated through a hippocampal-dependent behavioral test, i.e., the Morris Water Maze task. Of note, swimming speed was similar whatever group of mice considered, at each phase of this protocol (learning, relearning, probe test 1 and 2; Figure 2A–D). Across the five days of training, the latency to find the hidden platform significantly decreased, thus testifying that spatial learning occurred (Figure 2E). Further, no group difference was revealed, demonstrating similar spatial learning performances. Then, when assessed 72 h after the last day of learning, spatial memory performances appeared preserved in SR-KO mice. Indeed, not only above the chance value of 25%, the percentage of time spent in the target quadrant during the probe test for SR-KO did not differ from WT mice (Figure 2F). These results suggest that reference memory is not affected when the SR gene is deleted.

The location of the platform was then moved from location for the relearning phase, which thus allowed the assessment of cognitive flexibility in SR-KO mice. Again, whatever the group considered, a significant decrease in the latency to find the platform was observed during the five days of relearning, without any difference between genotypes (Figure 2G). Besides, the two groups displayed significant flexibility of memory performances (time spent in the new target quadrant during the probe test significantly above the chance value), without any between-genotype difference (Figure 2H).

### 2.2. SR Deletion Does Not Change the Functional Output of CA1 Hippocampal Networks

To investigate the basal excitatory synaptic transmission, extracellular field recordings were conducted and synaptic efficacy (I_SE_) was determined in control of aCSF for three increasing intensities of stimulation (see Section 4. Material and Methods). No significant alteration of I_SE_ was found in SR-KO mice regardless of the stimulation intensity (Figure 3A), confirming that the basal synaptic transmission is not affected by the SR deletion (see also [10,23,28]). Also, the paired-pulse facilitation ratio (PPF) was identical in SR-KO and WT mice, indicating that the probability of presynaptic glutamate release is not altered as well (Figure 3B).

As illustrated in Figure 3C, a 100 Hz tetanic stimulation-induced long-lasting potentiation was recorded both in WT and SR-KO mice. A similar magnitude of LTP was observed, regardless of the genotype considered (Figure 3D). To go further and because NMDAR activation is critical for LTP expression, NMDAR-mediated synaptic potentials induced at low-frequency stimulation were then pharmacologically isolated (see Material and Methods, Figure 3E), and possible changes of I_SE_ in these conditions were investigated in SR-KO mice. However, no significant difference between the two experimental groups was found regardless of the stimulus intensity (Figure 3F), indicating that NMDAR activation in CA1 networks is preserved in SR-deleted mice.

### 2.3. Changes in Glycine Levels Contribute to Preserving Functional Plasticity in SR-KO Mice

To uncover possible mechanisms allowing preserved memory capacities together with intact LTP expression in SR-KO mice, HPLC analysis of amino acids levels was first performed within hippocampi of both genotypes of mice (Appendix A). In agreement with previous studies [26,29], hippocampal d-serine levels were drastically reduced by 95% in transgenic mice compared to WT mice (Figure 4A). Furthermore, we found that the incubation of hippocampal slices with the d-serine deaminase to deplete d-serine levels [30] significantly reduced LTP magnitude in WT mice, but was without any effect in slices from SR-KO mice (Figure 4B). The data indicate that d-serine mediates NMDAR-dependent LTP in WT mice but that SR-KO may use another mechanism to preserve LTP. Since glycine also contributes to LTP expression at CA3/CA1 hippocampal connections of adult mice ([7,28]), we hypothesized that possible changes in the levels of glycine may compensate for the loss of d-serine at synapses, thus preventing a decrease in LTP expression in SR-KO mice. In agreement with this possibility, we found that glycine levels were significantly increased in the hippocampus of SR-KO mice when compared to WT littermates (Figure 4A).

### 2.4. Serine Racemase Deletion Changes the Excitation/Inhibition Balance of CA1 Hippocampal Networks

Patch-clamp recordings carried out on isolated excitatory glutamatergic transmission demonstrated impairments on LTP in SR-KO [23,26]. On the other hand, extracellular recordings analyzing the overall synaptic activity of the neuronal networks did not show any LTP changes [10,28]. Beyond evident methodological considerations, these two electrophysiological approaches differ in their consideration (or not) of close environmental neuronal networks. We therefore investigated if possible changes in the E/I balance in SR-KO mice could account for these apparent discrepancies. In order to investigate whether inhibitory tone is affected in SR-KO mice, we monitored the effects of the GABA_A_ receptor antagonist bicuculline (10 µM) on I_SE_ of NMDAR-mediated fEPSPs.

First of all, it worth mentioning that compared to their corresponding control group, NMDAR activation was increased for all stimulation intensities in the presence of bicuculline whatever the genotype considered (Figure 5A). However, the increase level induced by bicuculline was almost twice weaker in SR-KO compared to WT mice and this was found regardless of stimulation intensity (mean of ~30% increase versus 15% for WT and SR-KO mice, respectively, 15 slices in each group). Hence, this result suggests that GABA inhibitory tone is altered (lower) in SR-KO mice. Indeed, although not significantly different, I_SE_ of NMDAR-mediated responses was weaker in SR-KO mice compared to WT under condition of blockade of the GABAR activity by bicuculline (Figure 5B). In addition, LTP magnitude was reduced in slices from SR-KO mice compared to WT littermates in the presence of bicuculline. This reduction was observed over the whole time-course including the first 15 min after the conditioning stimulation (corresponding to short-term potentiation) and the last 15 min of recording (corresponding to LTP) (Figure 5C).

## 3. Discussion

This study aimed at re-evaluating the functional plasticity in CA1 networks of hippocampal slices and the related cognitive abilities in mice whose SR gene was invalidated. HPLC analysis confirmed the loss of d-serine in the hippocampus of SR-KO mice [26,29] and showed that it was associated with a small but significant increase in glycine levels. This compensatory response could explain why NMDAR activation and LTP expression in extracellular recordings are not affected in SR-KO mice and why memory abilities are preserved. However, functional deficits are recorded in hippocampal slices from the SR-KO mice when the intrinsic GABA activity is pharmacologically blocked with bicuculline, indicating that a change in the E/I balance could also contribute to maintaining functional plasticity of the CA1 hippocampal networks in the absence of d-serine.

We first confirmed that the basal glutamate synaptic transmission is preserved in SR-KO mice as revealed by the lack of alteration in the probability of presynaptic glutamate release using the electrophysiological PPF protocol (see also [10,23,28]). Besides, we observed that LTP expression and NMDAR activation in SR-KO mice were not affected in control conditions, but were reduced when the GABA_A_ receptor activity was pharmacologically suppressed. Our observations therefore suggest a weaker inhibitory tone in the neuronal networks after the genetic suppression of SR. In agreement, reduced frequency of spontaneous inhibitory postsynaptic currents was recently reported in CA1 pyramidal cells of SR-KO mice [31], indicating a weaker density of inhibitory synapses, as also supported by a decrease in levels of the vesicular GABA transporter in those animals [31]. We now raise the possibility that the loss of d-serine in SR-KO mice could also induce a weaker activation of NMDARs present on GABAergic interneurons themselves, thus reducing their inhibitory function. This point remains to be investigated using patch-clamp recordings of inhibitory interneurons. Importantly, the change in the E/I balance associated with the loss of d-serine could explain why LTP expression is not affected in our studies evaluating the overall activity of the CA1 hippocampal networks (see also [10,28]), whereas it is impaired when inhibition is shut down (see [23,26]).

It is noteworthy that our behavioral investigations (locomotor activity, anxiety-like behaviors, spatial learning, reference memory, and cognitive flexibility) did not reveal major deficits of memory performances in SR-KO mice. Like in LTP, the absence of behavioral changes in learning and memory tests could be related to the modified E/I balance of the CA1 network preserving its overall activity. Nevertheless, SR-KO mice seem to exhibit some traits of behavioral modifications in stress-related situations. We observed a decrease in anxiety-related behavior (% time spent in the center during the first minute) in the open-field, while data from the literature report impairments in other behaviors such as contextual fear conditioning, chronic social defeat stress or socio-communicative behaviors [25,32,33,34]. These altered behavioral responses in SR-KO mice could be associated with the reduced inhibitory tone induced by d-serine removal. Indeed, modulation of GABA activity is known to alter several behaviors such as locomotor activity, anxiety-related behaviors, and memory [35,36,37,38,39,40].

A normal LTP in SR-KO mice under control conditions is surprising since the necessary NMDAR activation by the main NMDAR co-agonist is completely absent, indicating the occurrence of potent compensatory mechanisms. Accordingly, we found that LTP in SR-KO mice was not affected by depletion of d-serine by treatment with recombinant d-serine deaminase, while it was impaired by the same enzymatic treatment in WT mice (see also [10]). Glycine is the major co-agonist driving the LTP expression at CA3/CA1 synapses during development while it only contributes at mature connections together with d-serine [7,28]. Interestingly, we found that in parallel to the loss of d-serine, glycine levels were increased in SR-KO mice. Therefore, in addition to a change in E/I balance, higher availability of glycine represents another mechanism that allows LTP to be maintained in the absence of d-serine.

Together, these results highlight that genetically modified models must be studied carefully because compensatory mechanisms can mask changes in synapses by maintaining overall network activity. Nevertheless, SR-KO mice are still an excellent animal model to validate our understanding of mechanisms that are dependent on d-serine availability at the synapse, as recently shown for the role of the system A-type of glutamine transporters in terminating d-serine signaling [41] or the Asc1 subtype of neutral amino acid transporters in mediating d-serine release [42,43].

## 4. Material and Methods

### 4.1. Animals

Male mice aged 7–12 months were provided by local facility (Centre Universitaire de Ressources Biologiques, Université de Caen, Normandie, France) including wild type C57BL/6J mice (WT, *n* = 26 assigned to electrophysiological studies, and *n* = 19 assigned to behavioral and biochemical analyses), and constitutive SR-KO mice (*n* = 18 for electrophysiological studies and *n* = 16 for behavioral and biochemical analyses) on a background originally issued from J.T. Coyle’s lab colony (Belmont, MA, USA). Mice were housed in groups of 5 in polycarbonate cages in standard controlled conditions with a reversed 12 h light/dark cycle (light on at 7 pm), controlled temperature (22 ± 1 °C) and free access to food and water in the home cage. All experiments were carried out in accordance with the European Communities Council Directive (2010/63/EU) regarding the care and use of animals in experimental procedures (APAFIS#24317, 12 May 2020).

### 4.2. Determination of Brain Amino Acid Levels

After cervical dislocation and decapitation of mice, hippocampi were dissected. The samples were homogenized with 10 volumes of 5% TCA, quickly frozen at −80 °C and subsequently processed for HPLC analysis as previously described [44], except that α-aminoadipic acid was substituted for L-homocysteic acid as the internal standard.

### 4.3. Extracellular Electrophysiology

Mice were anesthetized with isoflurane and killed by decapitation. Hippocampi were rapidly dissected, and transverse hippocampal slices (400 µm) were prepared in ice-cold artificial cerebrospinal fluid (aCSF) and placed in a holding chamber for at least 60 min. The composition of aCSF was (in mM): NaCl 124, KCl 3.5, MgSO_4_ 1.5, CaCl_2_ 2.3, NaHCO_3_ 26.2, NaH_2_PO_4_ 1.2 and glucose 11 (pH 7.4). A single slice was transferred to the recording chamber and continuously submerged with aCSF pre-gassed with a 95% O_2_/5% CO_2_ mixture.

Extracellular recordings were obtained at room temperature from the apical dendritic layer of the CA1 area using glass micropipettes (2–5 MΩ) filled with 2 M NaCl. To assess basal neurotransmission, presynaptic fiber volleys (PFVs) and non-NMDAR-mediated field excitatory postsynaptic potentials (fEPSPs) were evoked at 0.1 Hz by electrical stimulation of Schaffer collaterals and commissural fibers located in the stratum radiatum. The averaged slope of three PFVs and fEPSPs was measured using Win LTP software [45,46]. To assess the level of receptor activation, an index of synaptic efficacy (I_SE_) corresponding to the fEPSP/PFV ratio was plotted against increasing stimulus intensity (300, 400 and 500 µA).

Paired-pulse facilitation (PPF) of basal synaptic transmission was induced by electrical stimulation of afferent fibers with paired-pulse (inter-stimulus interval of 30 ms). PPF was calculated as the ratio of the slope of the second stimulation to the first one.

Specific NMDAR-mediated fEPSPs were isolated in slices perfused with low Mg^2+^ (0.1 mM) containing aCSF and supplemented with the non-NMDAR antagonist 2,3-dioxo-6-nitro-1,2,3,4-tetrahydrobenzoquinoxaline-7-sulfonamide (NBQX, 10 µM). Again, I_SE_ was plotted against stimulus intensity (300, 400 and 500 µA) in these conditions to assess the level of NMDAR activation. Effects of exogenous application of the GABA_A_ receptor antagonist bicuculline (10 µM) were evaluated by comparing I_SE_ before and 15 min after the addition of the drug to the aCSF.

In experiments investigating high-frequency stimulation (HFS)-induced long-term potentiation (LTP) of synaptic transmission, a test stimulus (0.1 Hz) was adjusted to obtain an fEPSP with a baseline slope of 0.1 V/s before the delivery of the conditioning stimulation consisting of 1 burst of pulses at 100 Hz delivered for 1 s. Testing with single pulses was then resumed for 60 min to determine the level of stable LTP. In some experiments, LTP was monitored in slices pre-incubated for at least 90 min with 20 µg/mL purified recombinant *Escherichia coli* D-serine deaminase (DsdA) along to deplete D-serine content. The purified recombinant enzyme was prepared as described previously (30).

### 4.4. Behavioral Experiments

Behavioral experiments were performed on the following sequence: actimeter, black and white box, open-field, and Morris Water Maze. All animals were handled daily for several weeks before the tests began, and experiments were conducted during the active-dark phase of the cycle. For habituation to a new environment, mice were placed in the room at least 30 min before the beginning of each experimental set. Apparatuses were cleaned with alcohol (70%) between each assessment.

### 4.5. Actimeter

Spontaneous locomotion was evaluated using an actimeter device (Imetronic), which corresponds to a Plexiglas cage (20 × 12 × 17 cm) containing 6 infrared beams allowing detection and counting of horizontal and vertical movements. Each mouse was placed in this individual cage and locomotor activity was measured by recording the number of interruption of beams of the red light over a period of 30 min through an attached recording system to the actimeter. The greater the number of beam interruptions, the more intense the locomotor activity of the animals and the inverse was also true. The recording of horizontal movements corresponds to horizontal activity and vertical movements to vertical activity, i.e., the number of rearings.

### 4.6. Black and White Box

The black and white box (Bioseb) was used to test anxiety and is based on rodents’ innate preference for dark and confined spaces, as opposed to a brightly lit environment. The experimental apparatus consisted of two plexiglass compartments composed of a large (28 × 27 × 27 cm) white/light and a second small (17 × 27 × 27 cm) black/dark, which communicates through an open door. Each mouse was introduced into the white compartment, and the experiment started when the mouse entered the black compartment (maximal latency criterion 120 s). Then, the mouse was allowed to freely explore both compartments for 5 min. The latency to enter for the first time into the black compartment, the number of transitions between compartments and the time spent in the white compartment was recorded. Five SR-KO mice were excluded from all black and white box analyses, one did not validate the maximum latency criterion, and the other four were excluded because they were outliers (Rout test).

### 4.7. Open-Field

The open-field test was performed in a device (50 × 50 × 20 cm, white plexiglass, Viewpoint) with an infrared floor allowing to record the animals in the absence of light (0 lux). Each mouse was allowed to freely explore the device for a period of 30 min. General exploration was measured by calculating the total distance traveled in the apparatus. Two zones (periphery vs. center) were designated digitally, and the time spent in each was measured using the software Ethovision (Noldus™, Wageningen, Netherlands). The percentage of time spent in the center zone, serving as an index of anxiety-like behavior, was assessed in the total duration of the test and only in the first one minute.

### 4.8. Morris Water Maze

Spatial learning, reference memory, and cognitive flexibility were assessed using a circular pool (120 cm diameter × 40 cm height, Viewpoint) filled with opaque non-toxic water kept at 22–23 °C. The pool was virtually divided into four equal quadrants. Large visual cues were displayed on the walls of the testing room around the maze. First, the learning phase consisted of five consecutive days during which the mice were trained to localize a platform (11 cm diameter, PF) hidden at 0.5 cm below the water surface. Each day was comprised of four consecutive trials (maximal duration 60 s) performed with a randomized starting point and a 60 s inter-trial interval. If the mouse did not reach the PF within 60 s, it was slowly guided to it and allowed to stay for 15 s. Reference memory was assessed during probe-test 1 with a unique trial (60 s) performed 72 h after the last day of learning, without PF. One day later, a relearning phase was performed with the same design as the learning phase, but the PF was transferred to the opposite quadrant. Once again, 72 h after the last day of relearning, probe-test 2 was performed by a unique trial (60 s). Each session was videotaped, and parameters (latency to find the PF, swimming velocity in learning/relearning phases, and time spent in each quadrant, swimming velocity in probe tests) were analyzed through the Ethovision software ((Noldus™, Wageningen, Netherlands).

### 4.9. Data Analysis

All data are expressed as mean ± S.E.M, except the dosage of amino acids, the number of transitions in the black and white box, and the percentage of time in the center in the open field, which are expressed in median ± interquartile. After the assessment of normality of the data distribution, unpaired or paired t-test, one-sample *t*-test, one or two-way ANOVA (with repeated measures or not), and Mann–Whitney were used accordingly. One- or two-way ANOVA with repeated measurements were followed by Bonferroni–Dunn post hoc tests for multiple comparisons. In all cases, differences were considered significant when *p* < 0.05.

Analyses were performed with GraphPad Prism 8.0 (GraphPad Software, San Diego, CA, USA) and Statview 5.0 (BrainPower Inc, Fremont, CA, USA).

## Figures and Tables

**Figure 1 ijms-21-09447-f001:**
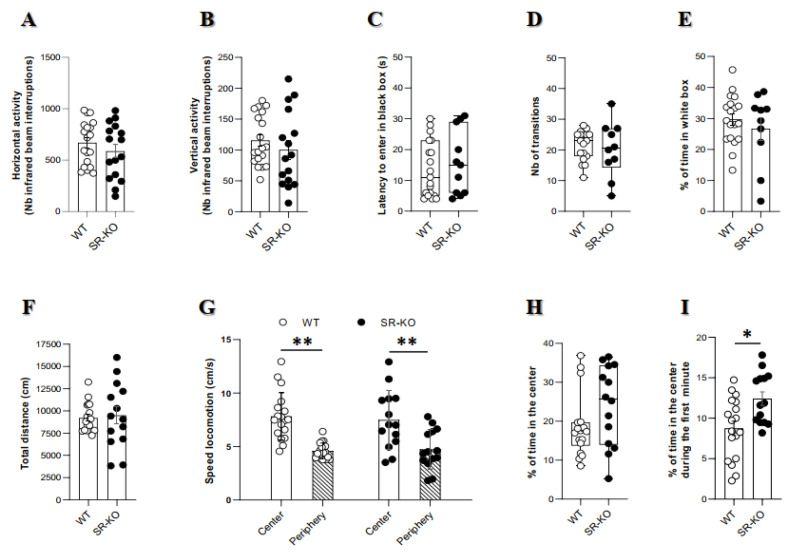
Spontaneous locomotion and anxiety-related behaviors are not consistently impacted by serine racemase (SR) deletion. (**A**,**B**). Horizontal (**A**) and vertical (**B**) activities in the actimeter between wildtype (WT) (*n* = 19) and SR-knockout (SR-KO) (*n* = 16) mice indicating unaffected general locomotor activity (unpaired t test with Welch’s correction). (**C**–**E**). Latency for first entry into the black box (**C**), number of transitions between black and white boxes (**D**) percentage of time spent in the white compartment, (**E**) in the black and white box of WT (*n* = 19) and SR-KO (*n* = 11) mice (unpaired *t* test with Welch’s correction for the latency and the percentage of time, Mann–Whitney test for the number of transitions). (**F**–**I**). Total distance traveled (**F**), speed locomotion in the center and the periphery (**G**), and percentage of time spent in the center during the 30 min period (**H**) or during the first one minute (**I**) in the open-field test of WT (*n* = 18) and, SR-KO (*n* = 14) mice. Except for a significant increase in the percentage of time in the center during the first minute, no behavioral changes were noted (* *p* < 0.05 for unpaired t test with Welch’s correction for the total distance and the percentage of time in the center in the first time, ** *p* < 0.001 for two-way ANOVA for the speed locomotion, Mann–Whitney test for the percentage of time in the center during all time of test).

**Figure 2 ijms-21-09447-f002:**
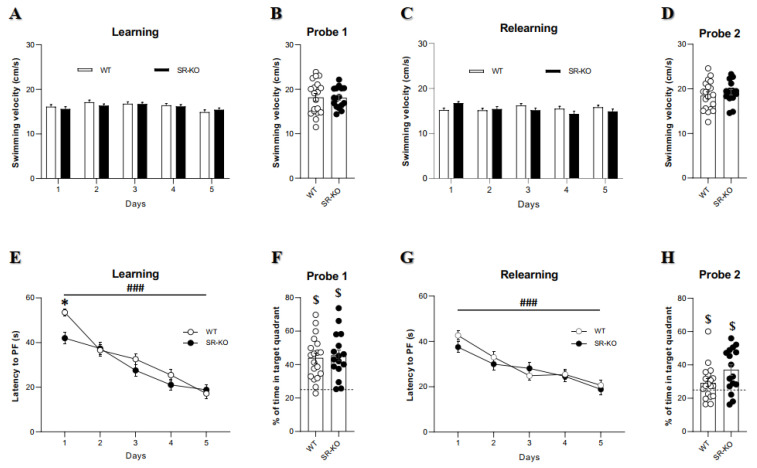
SR deletion does not impair spatial learning, reference memory, and cognitive flexibility in the Morris water maze (MWM). (**A**–**D**). Swimming velocity during the learning, probe test 1, relearning and the probe test 2 in WT control and SR-KO mice (two-way ANOVA) (**E**). Latency to reach the platform (PF) during the spatial learning (### for *p* < 0.001 for two-way ANOVA repeated measures). The latency was lower during the first day of learning in SR-KO mice compared to WT mice but not on the other days (* *p* < 0.05 with two-way ANOVA). (**F**). Percentage of time spent in the target quadrant during the probe test 1 in WT and SR-KO mice. The percentages were significantly above the chance value of 25% shown by the dotted line and no difference between groups ($, *p* < 0.05 one-sample *t*-test vs. >25, unpaired *t*-test with Welch’s correction). (**G**). Latency to reach the PF during the relearning in WT and SR-KO mice. Both groups displayed a significant decrease in latency to find the PF across the five days, and there was no difference between SR-KO and WT mice for each day (### *p* < 0.001 for two-way ANOVA repeated measures). (**H**). Percentage of time spent in the new target quadrant during the probe 2 in WT and SR-KO. Both groups displayed a percentage significantly above the chance value of 25% shown by the dotted line ($ *p* < 0.05 one-sample *t*-test vs. >25, unpaired *t*-test with Welch’s correction). A number of 19 WT mice and 16 SRKO mice were tested in every stage of the MWM.

**Figure 3 ijms-21-09447-f003:**
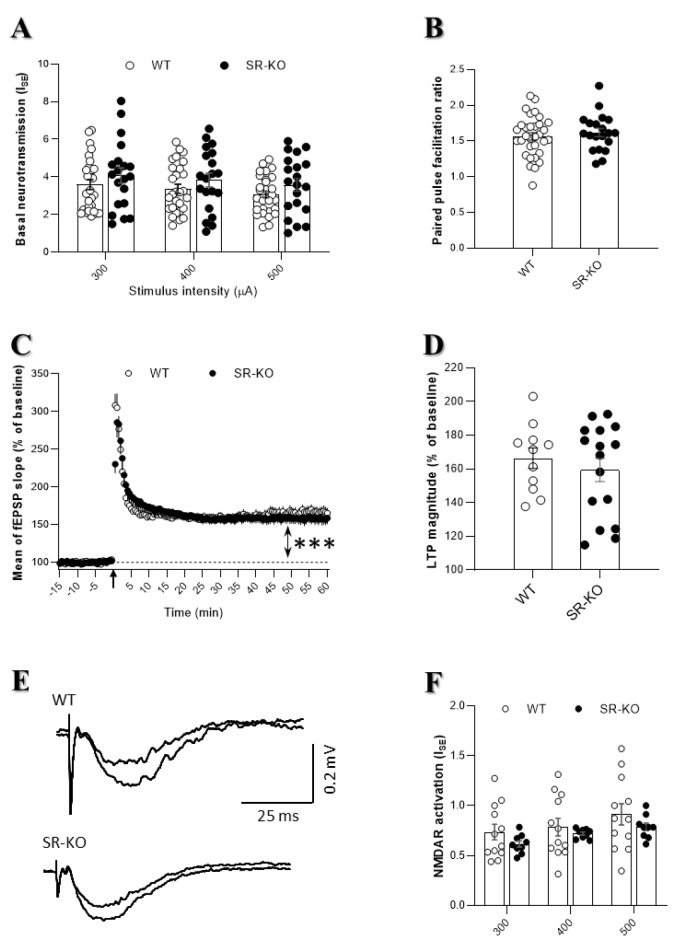
SR deletion does not change the functional output of CA1 hippocampal networks. (**A**). Index of synaptic efficacy (I_SE_) corresponding to the fEPSP/PFV slope ratio of AMPA receptor-mediated synaptic responses plotted against current intensity in WT (*n* = 28 slices) and SR-KO mice (*n* = 20 slices) (using two-way ANOVA). (**B**). Paired-pulse facilitation ratio of synaptic transmission in the WT (*n* = 32 slices) and SR-KO (*n* = 20 slices) mice (unpaired t-test with Welch’s correction). (**C**). Time-course of long-term potentiation of synaptic transmission induced by a conditioning stimulus of 1 × 100 Hz (black arrow) in WT (*n* = 11 slices) and SR-KO (*n* = 16 slices) mice. LTP is promoted in both groups, in regard to the last 15 min compared to baseline (*** *p* < 0.0001 for one sample t-test vs. 100 for each group). (**D**). LTP magnitude during the last 15 min in WT and SR-KO mice (two-way ANOVA repeated measures). (**E**). Superimposed traces of long-lasting N-methyl-D-aspartate (NMDAR)-induced fEPSPs recorded in a slice from a WT (up) and SR-KO mouse (down) at 300 and 400 µA stimulation intensities. (**F**). I_SE_ of NMDAR-mediated synaptic potentials isolated in WT (*n* = 13 slices) and SR-KO (*n* = 9 slices) mice (two-way ANOVA).

**Figure 4 ijms-21-09447-f004:**
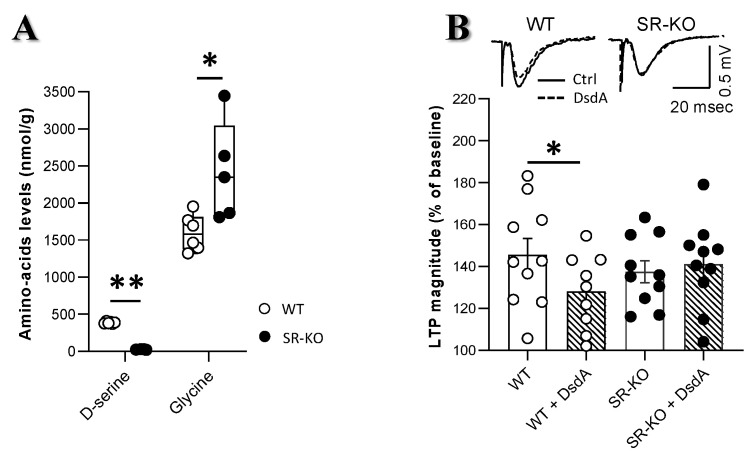
Changes in glycine levels contribute to preserving functional plasticity in SR-KO mice. (**A**). Comparison of d-serine and glycine levels (in nmol/g of hippocampal tissues) determined in WT (*n* = 6) and SR-KO (*n* = 5) mice. d-serine was mostly absent in the transgenic mice while glycine was significantly elevated (* *p* < 0.05 and ** *p* < 0.001 for Mann–Whitney test). (**B**). Superimposed traces of fEPSP slope after HFS stimulation in a slice from a WT (left) and SR-KO mouse (right) with or without d-serine deaminase (DsdA) respectively dashed line and full line. Comparison of mean LTP magnitude during the last 15 min in WT and SR-KO mice recorded in control aCSF (*n* = 10 slices in both groups) and slices incubated with the DsdA to deplete d-serine levels (*n* = 9 and *n* = 10 slices, respectively). LTP magnitude was significantly reduced by the d-serine depletion in WT mice while remained unaffected in SR-KO mice (two-way ANOVA; * *p* < 0.05 for paired *t* test in the last 15 min).

**Figure 5 ijms-21-09447-f005:**
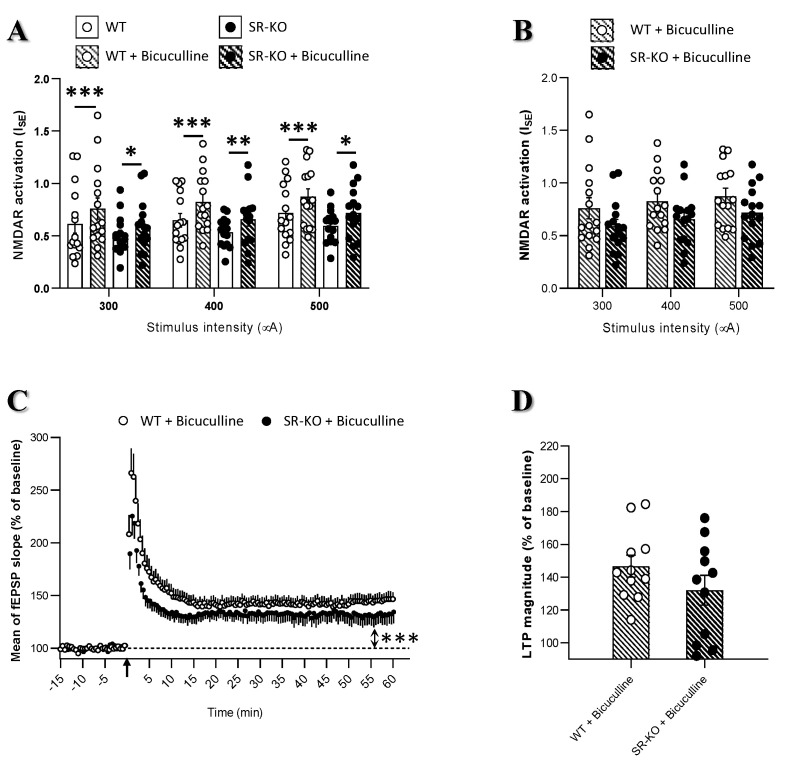
Serine racemase deletion changes the excitation/inhibition balance in CA1 hippocampal networks. (**A**). Effects of the GABA_A_ receptor antagonist bicuculline (10 µM) on I_SE_ of NMDAR-mediated fEPSPs. Increase in NMDAR activation occurred in both experimental groups at all stimulation intensities but the significance was much lower in SR-KO (*n* = 15 slices) than in WT mice (*n* = 15 slices) (* *p* < 0.05, ** *p* < 0.001, *** *p* < 0.0001 for paired *t*-tests). (**B**). Comparison I_SE_ of NMDAR-mediated synaptic potentials in WT and SR-KO mice after blockade of the GABA_A_ receptor activity. (**C**). Time-course of LTP expression in WT (*n* = 15 slices) and SR-KO (*n* = 15 slices) mice in the presence of bicuculline (10 µM) (*** *p* < 0.0001 for one sample *t*-test vs. baseline for each experimental group). (**D**). Comparison of mean LTP magnitude calculated for the last 15 min in WT and SR-KO mice in slices treated with bicuculline showing weaker potentiation in the SR-KO mice.

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
