# Peer review of "Serine Racemase Deletion Affects the Excitatory/Inhibitory Balance of the Hippocampal CA1 Network"

_ijms, 2020, doi:10.3390/ijms21249447_

Round 1
Reviewer 1 Report
The manuscript by Ploux et al. looks for phenotypic changes at the behavioral level and in the hippocampal circuit in animals deprived of serine racemase (SR). The expectation of these studies was to find changes in the NMDA-R physiology in these KO animals that could lead to substantial effects on synaptic plasticity and behavioral assays related to the hippocampus.
While this study could be of potential interest, given that D-serine is a co-agonist of NMDA-R, the results presented are incremental. In addition, there are some significant weaknesses listed here:
-Figure 4 lacks rigor as it has no representative traces; data should be compared using ANOVA not t-tests
-Figure 5 shows a very small phenotype at 15 min after induction of LTP, while not commenting on reduction of STP
-Using only one pattern stimulation at one frequency is not sufficient to claim a robust change in E/I tone
-More studies addressing mechanisms are required in order to propose a role of D-serine in regulating the E/I tone
-The lack of changes in behavior decreases the entusiasm for the studies making the reader wondering what is the relevance of the changes found in the E/I tone.
Author Response
Point 1-Figure 4 lacks rigor as it has no representative traces; data should be compared using ANOVA not t-tests
We thank the referee for this accurate comment. According to the suggestion, representative traces of fEPSPs recorded in WTs and SR-KO mice 60 min after the conditioning stimulation in control and DsdA conditions have been added in the figure 4B. As regards to statistical test used, no overall effect was found between four groups with the two-way ANOVA. Thereafter, while we focused the statistical analysis on the effect of DsdA (a priori hypothesis), we reported a statistical difference for WT groups of mice.
Point 2-Figure 5 shows a very small phenotype at 15 min after induction of LTP, while not commenting on reduction of STP
We do thank the reviewer for her/his valuable comment. In fact, we also noticed that – only under condition of bicuculline application, SR-KO mice displayed a lower Short-Term Potentiation (first 15min following induction of the conditioning stimulus) compared to WT. However, behavioral changes are commonly related in the literature to change in LTP and hence did not mention this difference of STP pattern. Of note, this difference was not statistically significant (two-way ANOVA, p=0.08). Nevertheless, the difference of STP pattern support the idea that the change in the inhibitory tone could affect the magnitude of the entire expression of the long-lasting synaptic plasticity in SR-KO mice. Short discussion on this observation is now included in the revised form of the manuscript (lines 199-202).
Point 3-Using only one pattern stimulation at one frequency is not sufficient to claim a robust change in E/I tone
We do agree with the referee. In fact, the prime aim of our manuscript was to reconciliate discrepancies of published data within the literature using this single stimulation protocol of LTP induction. Afterwards, while writing of the manuscript, the notion of a change in E/I tone emerged as a relevant working hypothesis, notably given the weaker responsiveness of low-frequency induced NMDAR synaptic potentials to GABAR blockade in SR-KO mice. Nonetheless, another stimuli conditioning should be tested prior to validate the E/I imbalance hypothesis. For instance, investigation of theta-burst stimulation, which is more prone to inhibitory regulation, appears relevant for our purpose. Therefore, we have moderated the strength of our comments (and suggest further experiments) in the conclusion of the article.
Point 4-More studies addressing mechanisms are required in order to propose a role of D-serine in regulating the E/I tone
In line with the previous comment, we fully agree that additional evidences are required to confirm our hypothesis. In fact, we had planned to look for possible changes in GABA levels using HPLC analysis as soon it will be possible to manipulate (COVID19). Of note, in the recent study of Jami et al, 2020, while using recordings of IPSCs in patch-clamped CA1 pyramidal cells, authors also argued for an altered E/I balance in SR-KO mice.
Point 5-The lack of changes in behavior decreases the enthusiasm for the studies making the reader wondering what is the relevance of the changes found in the E/I tone.
We do not share the lack of enthusiasm of the reviewer for our behavioral data. Indeed, all the contrary, we were quite excited to realize that even if our electrophysiological data claim for E/I imbalance in SR-KO mice, no major behavior deficit was noticeable. In fact, what really struggle our mind is that compensatory mechanisms at work might however be overpassed in very stressful situations. This may hence explain why – according to the level of stress of behavioral test performed – some authors (but not all) may observe behavioral deficits in SR-KO mice.
Reviewer 2 Report
The study by Ploux and colleagues aim to understand how bicuculline affects neuronal activity in the CA1 region of the hippocampus in SR-KO mice. I think the study and manuscript are decent. Comments and concerns are outlined below.
There is little difference between the WT and SR-KO mice on almost all the behavior tasks and recordings. The main difference that the authors find is that there is a difference within groups when compared to baseline. I think the actual main point of the paper is in the first sentence of the last paragraph of the discussion, "Together, these results highlight that genetically modified models must be studied carefully because compensatory mechanisms can mask changes in synapses by maintaining overall network activity." Which is a true statement, but the authors spend a lot of time talking about differences in SR-KO mice compared to WT and we don't see any of these differences emerge in the data. I think the comparisons between WT and SR-KO should be toned down throughout the manuscript.
Throughout the discussion, the authors say that SR-KO mice display reduced inhibitory tone. I'm not sure that the manipulations in the study support this claim. Regardless, it would be of significant interest for this paper for the authors to measure GABA (in one of a variety of ways) in the hippocampus of SR-KO mice at baseline.
The legend for figure 5A is odd, it's difficult to distinguish which bars represent which groups.
For the black and white box task, there were four mice thrown out of the SR-KO group because they were outliers....that is a lot of outliers for a single task. Especially when analyzing this with Grubbs test. Is this correct?
The manuscript needs significant revisions in language and grammar.
Author Response
Point 1-There is little difference between the WT and SR-KO mice on almost all the behavior tasks and recordings. The main difference that the authors find is that there is a difference within groups when compared to baseline. I think the actual main point of the paper is in the first sentence of the last paragraph of the discussion, "Together, these results highlight that genetically modified models must be studied carefully because compensatory mechanisms can mask changes in synapses by maintaining overall network activity." Which is a true statement, but the authors spend a lot of time talking about differences in SR-KO mice compared to WT and we don't see any of these differences emerge in the data. I think the comparisons between WT and SR-KO should be toned down throughout the manuscript.
We totally agree with this overall observation of the referee and particularly when she/he points out our conclusion of the importance of compensatory mechanisms in genetically modified animals. However, it must be kept in mind that one initial aim of our investigation was to understand some discrepancies found in the literature data regarding the preservation (or not) of functional expression of CA1 neuronal networks following D-serine deprivation. In this purpose, we think that the absence of salient differences in WTs and SR-KO is an important result because it is at the origin for the search of compensatory mechanisms.
Point 2-Throughout the discussion, the authors say that SR-KO mice display reduced inhibitory tone. I'm not sure that the manipulations in the study support this claim. Regardless, it would be of significant interest for this paper for the authors to measure GABA (in one of a variety of ways) in the hippocampus of SR-KO mice at baseline.
We fully agree with the referee for this point that has already been considered in the comments of the referee 1. Indeed, GABA levels quantification in SR-KO would help to characterize a change in inhibitory tone in those mice although tissue levels do not predict of alterations of release mechanisms. Nevertheless, as previously stated, the current health situation in France does not allow to perform the required HPLC analyses that will be done as soon as possible. We also explain above that even incremental, our results complement those of Jamy (2020) very recently reported in SR-KO (including a reduced frequency of spontaneous inhibitory postsynaptic currents, a lower density of inhibitory synapses, decreased levels of the GABA vesicular transporter) that reinforces the conclusion of an impaired GABA inhibition in the absence of D-serine.
Point 3-The legend for figure 5A is odd, it's difficult to distinguish which bars represent which groups.
We do apologize for this. Modifications have been made accordingly.
Point 4-For the black and white box task, there were four mice thrown out of the SR-KO group because they were outliers....that is a lot of outliers for a single task. Especially when analyzing this with Grubbs test. Is this correct?
Indeed, we do apologize for the confusing sentence, all outliers have been identified by the ROUT test (not limited to detection of only one outlier like Grubbs test). This has now been corrected in the new manuscript. Of note, the high number of outliers in SR-KO mice should also be considered with regard to level of stress induced by the behavioral test, reinforcing idea of overwhelmed compensatory mechanisms in stressful condition.
Point 5-The manuscript needs significant revisions in language and grammar.
This article has been checked by a native English speaker.